



# Elasticity curves describe streamflow sensitivity to precipitation across the entire flow distribution

Bailey J. Anderson[1], Manuela I. Brunner[2, 3], Louise J. Slater[1], and Simon J. Dadson[1, 4]

[1]School of Geography and the Environment, University of Oxford, Oxford, UK
[2]Institute for Atmospheric and Climate Science, ETH Zurich, Zurich, Switzerland
[3]Institute for Snow and Avalanche Research SLF, Swiss Federal Institute for Forest, Snow and Landscape Research WSL, Davos, Switzerland
[4]NERC Centre for Ecology and Hydrology, Wallingford, UK

**Correspondence:** Bailey J. Anderson (bailey.anderson@ouce.ox.ac.uk)

**Abstract.** Streamflow elasticity is a simple approximation of how responsive a river is to precipitation. It is represented as a ratio of the expected percentage change in streamflow for a 1% change in precipitation. Typically estimated for the annual median streamflow, we here propose a new concept in which streamflow elasticity is estimated across the full range of streamflow percentiles in a large-sample context. This "elasticity curve" can be used to develop a more complete depiction of how

streamflow responds to precipitation. We find three different elasticity curve types which characterize this relationship at the annual and seasonal timescales in the USA, based on two statistical modelling approaches, a panel regression which facilitates causal inference and a single catchment model which allows for consideration of static attributes. Type A describes catchments where low flows are the least and high flows are the most responsive to precipitation. The majority of catchments at the annual, winter, and fall timescales exhibit this behavior. Type B describes catchments where the response is relatively consistent across

the flow distribution. At the seasonal timescale, many catchments experience a consistent level of response across the flow regime. This is especially true in snow-fed catchments during cold months, when the actual elasticity skews towards zero for all flow percentiles while precipitation is held in storage. Consistent response is also seen across the majority of the country during spring when streamflow is comparatively stable and in summer when evaporation demand is high and soil moisture is low. Finally, Type C describes catchments where low flows are the most responsive to precipitation change. These catchments

are dominated by highly flashy low flow behavior. We show that the curve type varies separately from the magnitude of the elasticity. Finally, we demonstrate that available water storage is likely the key control which determines curve type.

## 1 Introduction

The relationship between streamflow and meteorological variables such as precipitation, temperature, and evaporation is often represented simplistically and may be poorly understood. Observations can provide better insight into assumed physical rela-

tionships. One data-based approach for quantifying the relationship between streamflow and precipitation, and for estimating future changes in streamflow, is the concept of "elasticity". Streamflow elasticity describes the sensitivity of streamflow to changes (relative to the long-term mean of the relevant time series) in any given climatic variables and is defined most fre-





quently as the percent change in the annual water balance or mean annual streamflow, resulting from a one percent change in a variable of interest (Schaake, 1990).

In simplistic terms, at the seasonal to annual timescales, streamflow magnitude represents the aggregated components of precipitation, evapotranspiration, storage, including antecedent moisture conditions, and water use. Hence, mean streamflow elasticity to precipitation is unlikely to approximate unity. In other words, a one percent change in precipitation is unlikely to result in a one percent change in streamflow. Instead, changes in precipitation tend to be amplified in streamflow, and elasticity estimates are typically greater than one. Streamflow elasticity to precipitation has been reported on extensively at the annual

timescale (e.g. Chiew et al., 2006; Sankarasubramanian et al., 2001; Tang et al., 2020; Milly et al., 2018; Berghuijs et al., 2017; Tsai, 2017), and more recently, the aggregated multi-annual scales (Zhang et al., 2022). Reported values range between 0.75 and 2 depending on the region and methodology (Sankarasubramanian et al., 2001; Tsai, 2017; Allaire et al., 2015) and may differ for increases vs decreases in precipitation, for instance average streamflow in arid regions tends to be more sensitive to precipitation decreases than increases (Tang et al., 2019).

Storage and discharge depend on the physical and climatological characteristics of a catchment, meaning that elasticity may vary widely in different locations (Smakhtin, 2001). Previous research has shown that the magnitude of elasticity estimates varies in line with, for instance, snow fraction which is inversely associated with higher streamflow elasticity to precipitation (Sankarasubramanian et al., 2001; Chiew et al., 2006) or the runoff coefficient which is inversely related to the actual magnitude of streamflow (Chiew et al., 2006; Sawicz et al., 2011). Similarly, Cooper et al. (2018) found that seasonal low flows in the

western United States were negatively correlated with snowfall and drainage rate, exhibiting lower elasticity in snow dominated and slow draining catchments.

The flow regime of any river is composed of low, average, and high flow components. These are seasonally occurring phenomena which may or may not be stationary in time (Lins, 2012; Slater et al., 2020). The dominant sources of streamflow differ depending on the segment of the hydrograph which is considered. For instance, low flows or base flows in natural rivers

are typically the result of inflow from catchment storage sources, such as groundwater, lakes, or wetlands (Smakhtin, 2001). Meanwhile, high streamflow magnitudes are controlled, in large part, by precipitation events and antecedent soil moisture conditions (Ivancic and Shaw, 2015; Slater and Villarini, 2016). It stands to reason that different percentiles of streamflow at both the annual and seasonal timescales will therefore experience different elasticities to precipitation changes. In other words, a one percent increase in total annual precipitation may not have the same effect on the annual low flows as it will on median

or high streamflow.

In addition to elasticity of streamflow estimated for the mean or median flow, previous empirical analyses have also explored the sensitivity of hydrologic extremes to precipitation. For instance, the relationships of low flows to temperature and precipitation (Kormos et al., 2016; Bassiouni et al., 2016; Tsai, 2017) as well as potential evaporation (Cooper et al., 2018), and snow pack volume (Godsey et al., 2014) have been studied at the regional and seasonal scales. Similarly, sensitivity of flood

flows has been studied relative to precipitation (Prudhomme et al., 2013; Brunner et al., 2021), antecedent precipitation (Slater and Villarini, 2016), and temperature (Wasko and Sharma, 2017; Wasko et al., 2019; Wasko, 2021). However, in each of these





analyses, streamflow responsiveness to climate change is conceptualized as the estimated response of a singular subsection of the flow distribution, e.g. a flow percentile such as $Q_{95}$.

There is limited understanding of how streamflow elasticity to precipitation varies across the complete flow distribution.
This combined information could provide insights into the physical relationships between streamflow, climate, and the physiographic characteristics of a catchment. Further, while average elasticity is useful in planning for things like water supply, irrigation and energy related projects, abnormally high and low flows are associated with the greatest strain on hydrological systems (Nemec and Schaake, 1982). Improving understanding of how the streamflow-precipitation relationship varies across the flow distribution is important for water management, and deeper exploration of elasticity can highlight rivers and streams
which are likely to be the most susceptible to or which are undergoing substantial disturbance resulting from precipitation change and human impacts such as water abstraction.

Median elasticity ($\varepsilon_P^{50}$) is typically estimated around the long-term mean for the 50th percentile of streamflow for individual catchments using the approach proposed by Schaake (1990) and Sankarasubramanian et al. (2001) (Eq. 1):

$$\overline{\varepsilon_P^{50}} = \frac{\Delta Q_{50}}{\Delta P} \frac{\overline{P}}{\overline{Q}} \tag{1}$$

where $Q_{50}$ is the 50th percentile of annual daily runoff and P is the annual sum of catchment-averaged precipitation. Others have used the coefficients from multivariate regression models, such as generalized and ordinary least squares regression (Andréassian et al., 2016; Potter et al., 2011), the benefits of which include simultaneous estimation of sensitivity to potential evaporation and precipitation, accounting for co-variation in these phenomena and providing a more robust estimate of elasticity (Andréassian et al., 2016). Probabilistic statistical estimation of elasticity also enables straightforward calculation of confidence
intervals.

Understanding where and why physically disparate catchments exhibit similar behavior is an important tool for discerning the physics of hydrological systems using observed large sample data sets. Geographic proximity is one of the best predictors of hydrologic similarity (Sawicz et al., 2011; Patil and Stieglitz, 2012). Hydrologic similarity is further characterized by shared climatological and topographic characteristics between catchments, features which are likely to change slowly across space
(Patil and Stieglitz, 2012; Winter, 2001). Hydrologic behavior has been shown to have a lower degree of regional similarity for low flows as compared to relatively higher streamflow percentiles, likely because local geographic conditions have greater influence over low flow regimes (Patil and Stieglitz, 2011). However, a functional classification of mean annual hydrographs revealed a high degree of regional similarity across hydrologic regimes (Brunner et al., 2020).

Due to the high degrees of regional hydrologic similarity found in previous work, we posit that climate elasticity will
vary substantially, but systematically, across the distribution of streamflow and across space. We propose the use of a new concept, "elasticity curves", as a means to interpret hydrological responses to precipitation (Figure 1 A). Elasticity curves can be generated by estimating elasticity for a series of discrete percentiles of streamflow. The combination of these discrete point estimates will then form a curve which represents the variation in streamflow sensitivity to climate across the annual and seasonal streamflow distributions (Figure 1 B). We use a multivariate single catchment elasticity estimation approach, as





well as more robust panel regression models to estimate elasticity. Panel models are capable of controlling for a large portion
of omitted variable bias, allowing for a causal interpretation of regression results (Nichols, 2007; Hsiao, 1995; Croissant and
Millo, 2018), and have been shown the produce more reliable elasticity estimates than single catchment models (Bassiouni
et al., 2016; Anderson et al., 2022), although their application for the expicit estimation of elasticity is, to date, not very
widespread.

In this study, we generate streamflow elasticity to precipitation curves ($\varepsilon_{c,P}$) for 805 rivers in the United States using
statistical modelling and clustering approaches. We address the following questions:

   1. Does $\varepsilon_{c,P}$ shape vary systematically and predictably across catchments?
   2. What catchment attributes best explain between-catchment variation in $\varepsilon_{c,P}$ shape?

## 2   Methods

We estimate the elasticity of streamflow to changes in precipitation at every 5th percentile of annual and seasonal flow in 805
perennial U.S. rivers. Taken from the Geospatial Attributes of Gages for Evaluating Streamflow version II (GAGES II) data
set, selected catchments were minimally influenced by dam storage, defined as having less than 1 day of upstream dam storage
(Anderson et al., 2022; Blum et al., 2020; Hodgkins et al., 2019), calculated by dividing the total upstream dam storage by
the estimated catchment annual runoff (Falcone, 2017). Additionally, all retained catchments had at least 30 years of 95%
complete, consecutive daily streamflow data between 1981 and 2022. Finally, we removed all ephemeral rivers and streams,
defined as streamflow records having any 0 flow days.

### 2.1   Data

Catchment attributes, including total upstream dam storage and average annual runoff and watershed boundaries were taken
from the same source (Falcone, 2017). The daily streamflow time series were extracted for the period 1981–2020 from the
USGS using the *R* package dataRetrieval (DeCicco et al., 2022). Gridded monthly precipitation and temperature (4 km resolu-
tion) were extracted from the Oregon State PRISM project using the R package prism (Edmund and Bell, 2020). We estimated
average daily precipitation (mm/day) for each timescale from the monthly values within each catchment boundary. We cal-
culated average daily PET (mm/day) for each timescale in R using the Hamon equation (Hamon, 1963; Crevensten, 2012;
Lu et al., 2007) with monthly temperature as previously described and estimated solar radiation from latitude and Julian date
(Crevensten, 2012). While GAGES II (Falcone, 2017) includes PET estimates, also calculated using the Hamon equation, we
recalculated these values in order to accurately represent the time period of the analysis. The Hamon equation was used to retain
consistency with the GAGES II data set and because this method has been shown to perform well relative to other approaches,
despite its simple formulation (Lu et al., 2007). Annual values were calculated for the period between September and August
so that all dates for the annual and seasonal timescales fall into corresponding "years". The seasonal values were estimated for
winter (December, January, February), spring (March, April, May), summer (June, July, August) and fall (September, October,
November).





## 2.2 Single catchment models

In the first instance, we fit log linear models (lm) using the ordinary least squares estimator, to every 5th percentile of the annual and seasonal flow regimes from the minimum streamflow magnitude ($Q_0$) to the maximum ($Q_{100}$) for each historical streamflow record (Eq. 2).

$$\ln(Q_{i,t}^q) = \alpha_{i,t} + \varepsilon_P^q \ln(P_{i,t}) + \varepsilon_E^q \ln(E_{i,t}) + \eta_{i,t}^q \qquad (2)$$

where $\ln(Q_{i,t}^q)$ is the natural logarithm of a streamflow percentile (q) calculated for time period (t) for catchment (i), $\alpha_{i,t}$ is the intercept, $\ln(P_{i,t})$ is the logarithm of catchment averaged daily precipitation, and $\ln(E_{i,t})$ is the logarithm of catchment averaged daily potential evaporation. The point estimate of precipitation elasticity is represented by the regression coefficient: $\varepsilon_P^q$ and potential evaporation elasticity is represented by $\varepsilon_E^q$. The error term is $\eta_{i,t}^q$.

The elasticity curve $\varepsilon_{c,P}$ is simply the combination of the percentile specific point estimates of elasticity ($\varepsilon_P^q$). For visualization purposes, we linearly interpolate between the points.

Understanding the shape of the elasticity curve is important in order to assess the responsiveness of different streamflow percentiles to changes in precipitation within a given catchment area. We do not explicitly try to explain variation in actual magnitude of elasticity in this work because this has been done extensively in other literature. We aim, instead, to identify catchments with a similar elasticity behavior across streamflow quantiles, and therefore seek to cluster the curves based on their shape, rather than the magnitude of the elasticity estimates. To achieve this, we normalize the curves relative to the elasticity of the minimum streamflow in each timescale, by subtracting $\varepsilon_P^0$ from each of the $\varepsilon_P^q$ estimates.

We then use Ward's minimum variance method (Ward, 1963) for agglomerative hierarchical clustering in *R* to group the complete elasticity curves for the individual catchments into clusters with similar shapes. Hierarchical clustering methods were chosen because the results are reproducible and not influenced by initialization and local minima (Murtagh and Contreras, 2012). We used the Euclidean distance measure for clustering, and Ward's algorithm was selected because it had the highest agglomerative coefficient as compared to the complete linkage, single linkage, and UPGMA algorithms, indicating stronger clustering structure.

The number of clusters for each temporal scale were selected through visual inspection of the dendrograms, silhouette plots, and the gap statistics. We additionally performed a sensitivity analysis in which we fit 2, 3, 4, and 5 clusters to the data and examined the spatial distribution of the prospective clusters. This resulted in the selection of 3 clusters for the annual, winter, and summer timescales and 2 clusters each for the spring and fall timescales. We then determined cluster type based on the difference between the average elasticity of the minimum and maximum flow in a given period. The number of clusters were chosen so that the fewest clusters possible would be selected for each temporal scale while still capturing the general shapes of the $\varepsilon_{c,P}$s. In spring and fall additional clusters did not result in a more informative classification.





## 2.3 Causal model design

In order to further validate the elasticity estimates, we constructed a causal, fixed-effects, panel regression model (Eq. 3) for each timescale ($\varepsilon_{c,P}^{g,q}$). The panel models were designed to control for confounding variables, and the clusters established from

the lm results were included as interaction terms to help explain variation in elasticity curve shape. A confounding variable is an attribute of a catchment or group of catchments which could influence both the dependent variable and independent variable, causing a spurious association.

    Time-invariant confounders at the catchment scale are controlled for by the streamgage-specific intercept $\alpha_i$. At the timescale of this study (30-39 years of data per site), the majority of catchment specific confounding variables may be reasonably ex-

pected to be time-invariant, for instance, topography. While some land cover changes are likely over the time period, a minority of catchments are likely to have experienced large percentages of detectable land cover change, and, when considered jointly in a panel model, the effects of land cover changes on streamflow will be unsubstantial relative to climatic effects (Anderson et al., 2022). Other variables, such as temperature and actual evapotranspiration, are partially or fully considered through the calculation or inclusion of other variables. More complex formulations of the panel model, which explicitly included eco-

regions and/or a control for time varying confounders at the national scale were considered, however, the resulting curves were not substantially different from one another, and thus the simplest model (Eq. 3) is used.

$$\ln(Q_{i,t}^q) = \alpha_{i,t} + \beta_1 \ln(P_{i,t}) + \beta_2 \ln(E_{i,t}) + \boldsymbol{\varepsilon_P^{g,q}} \ln(P_{i,t})g_i + \varepsilon_E^{g,q} \ln(E_{i,t})g_i + \eta_{i,t}^q \tag{3}$$

    where $\ln(Q_{i,t}^q)$ is the natural logarithm of a streamflow percentile (q) calculated for time period (t) for catchment (i), $\alpha_{i,t}$ is the streamgage-specific intercept, $\ln(P_{i,t})$ is the logarithm of catchment averaged daily precipitation, and $\ln(E_{i,t})$ is the

logarithm of catchment averaged daily potential evaporation. The elasticity curve cluster for each catchment represented by a categorical variable (g), and $\ln(P_{i,t})g_i$ and $\ln(E_{i,t})g_i$ are interaction terms between the assigned cluster and precipitation or potential evaporation. Precipitation elasticity, the effect measured by this model, is represented by the regression coefficient: $\varepsilon_P^{g,q}$ and potential evaporation elasticity is represented by $\varepsilon_E^{g,q}$. The error term is $\eta_{i,t}^q$. Autocorrelation in fixed effects panel models can lead to the underestimation of standard errors. We address this concern by clustering standard errors at the stream-

gage level as in Anderson et al. (2022). The panel regression results are normalized following the same procedure as the lms – by subtracting $\varepsilon_P^{g,0}$ from each $\varepsilon_P^{g,q}$ value.

    Throughout the work the following notation will be used: elasticity point estimates are represented by $\varepsilon_P^q$ where $q$ may be replaced with a number 0-100 to represent the streamflow percentile, or an overbar may be used to indicate the average of all $\varepsilon_P^q$ values, ex. $\overline{\varepsilon_{c,P}}$; Elasticity curves are represented by $\varepsilon_{c,P}$ in general and by $\varepsilon_{c,P}^g$ when the panel model results are specifically

referenced.

## 2.4 Attribution of elasticity curve classification

Finally, we are concerned with the drivers behind variability in elasticity curve shape. Therefore, we consider explanatory variables which have previously been shown to be related to between-catchment variation in the magnitude of elasticity as





well as additional variables related to hydrologic responsiveness. These variables include: the slope of the flow duration curve
calculated for low flows (lowest third), average flows (middle third), and high flows (highest third), runoff coefficient, average
annual temperature, aridity index, mean elevation, slope and, drainage area, snow fraction, and average permeability and
latitude (Falcone, 2017). We additionally consider standard baseflow index (BFI) calculated over a time window of five days,
and a longer "delayed flow index" (DFI) calculated over a time window of 90 days as in Gnann et al. (2021). Our intention here
is to capture baseflow from different sources – BFI aims to separate event from inter-event flow and DFI aims at separating
seasonal variation from inter-annual baseflow (Gnann et al., 2021; Stoelzle et al., 2020). DFI has been previously shown
to be much more clearly related to geology as compared to BFI. The full equations and specifications for the explanatory
terms are included in the appendix. Finally, we consider six categorical seasonality variables: most important precipitation
season (winter, spring, summer, fall), calculated as the season in which the largest precipitation amount falls, least important
precipitation season, calculated as the season in which the least amount of precipitation falls, low flow season, and high flow
season. Further, we include combinations of most important precipitation season and low flow season, as well as least important
precipitation season and low flow season (ex. winter_summer, in the instance that winter is the most important precipitation
season and summer is the most important flow season). These final two seasonality metrics are intended to shed light on
whether streamflow is in phase with precipitation.

To attribute the drivers of between-catchment variation in elasticity shape, we use random forest regression models for
multivariate estimation of variable importance for the prediction of cluster membership in each temporal scale. The clusters
are frequently imbalanced in terms of the number of sites in each group, so we train the model on a sub-sample of the data set
which consists of 80% of the sites in the smallest cluster and equivalent quantities of each additional cluster randomly selected
from the complete data set. We then test the model performance using a sample which consists of the remaining 20% of the
smallest cluster, and quantitatively equivalent samples of each additional cluster. We repeat the random sampling and model
fitting process 10 times per temporal scale and then calculate the average actual accuracy across the 10 iterations.

## 3   Results

Figure 2 (A and C) shows the average normalized elasticity curves for each temporal scale resulting from both modelling
approaches. The normalized curves have been clustered so that catchments with similar curve shapes are in the same group.
The curves in panel A were produced using linear regression models fit to each catchment individually (Eq. 2), then averaged
for each cluster and plotted with the interquartile range of the respective $\varepsilon_p^q$ values for every catchment in that cluster. We use
the interquartile range because the lms result in a distribution of $\varepsilon_p^q$ values for each streamflow percentile (one per site) and the
resultant curve is an average of all sites in a cluster. The curves in panel C were produced using the panel regression approach
(Eq. 3) and are plotted with the normalized 95% confidence intervals of the panel model. The panel regression model results
in one $\varepsilon_p^{q,g}$ value for each percentile, and allows for the calculation of a confidence interval. These results are strikingly similar
to those of the single catchment regression models, increasing confidence in the overall procedure.





We find three main curve types which we define as: curve type A - where the cluster average curve is positively sloping and the difference between $\varepsilon_P^0$ and the largest point estimate in the average curve is greater than 0.75 percentage points; curve type B - where the cluster average curve is relatively flat and the absolute difference between these points falls between -0.75 and 0.75 percentage points; and curve type C - where the cluster average curve is negatively sloping and the difference between
$\varepsilon_P^0$ and the largest point estimate of the average curve is less than -0.75 percentage points. We further define two sub-types of curve types A and C: "strong" with greater than a 1.25 percentage point difference between $\varepsilon_P^0$ and the largest point estimate and "weak" (0.75 - 1.25 percentage points). This division is merely a heuristic for separating the clusters. Some catchments within each type class have total absolute differences in elasticity estimates which do not comply with this division.

At the annual timescale, 91% of catchments exhibited type A curves, demonstrating that in an overwhelming majority of
cases larger streamflow quantiles are proportionally more responsive to precipitation. Of these, 31% (251 catchments) were grouped into a single class for which the average $\varepsilon_{c,P}$ has a strongly positive slope (curve type A: strong), and 60% of catchments (495) were clustered into a weakly positive class (type A: weak). In catchments with curve type A, where $\varepsilon_{c,P}$ has a positive slope, higher streamflow percentiles are increasingly more responsive to a one percent change in precipitation than are low flows. Some catchments, predominantly in the eastern portion of the country, exhibit different behavior. 7%
of catchments (58 catchments) were clustered into a group with strongly negative $\varepsilon_{c,P}$ (curve type C: strong). A negatively sloping elasticity curve shape indicates that high flows are relatively less responsive to precipitation variation than are lower flows. In other words, variation in precipitation predominantly effects the hydrologic response of larger streamflow percentiles for catchments with a positively sloping $\varepsilon_{c,P}$, and lower streamflow percentiles in catchments with negatively sloping $\varepsilon_{c,P}$.

In winter, fall, and spring, none of the cluster-average elasticity curves are negatively sloping. 31% of catchments (246) in the
fall, 26% (211) in winter, and 65% (524) in spring are grouped into a cluster for which $\varepsilon_{c,P}$ can be described as relatively flat (curve type B), defined here as having a range of normalized $\varepsilon_P^q$ values between -0.75 and positive 0.75. In winter, catchments with curve type B are mostly concentrated at high latitudes and mountainous regions, while in the fall, these catchments are geographically more widespread, existing both in the north, the southwest, and to some extent, throughout the gulf coast. A flat elasticity curve denotes a catchment in which the responsiveness of streamflow to changes in precipitation is consistent
across the distribution. The remaining clusters are positively sloping curves. Similarly, 78% (626) of catchments in the summer season are curve type B. Meanwhile 111 catchments are curve type A (strongly positive), and a cluster with the remaining 8% (68) of catchments is generally negatively sloping (type C: weak). Finally, 281 catchments in spring are weakly positive (type A: weak). In spring, the absolute difference between the cluster-specific $\varepsilon_P^0$ and $\varepsilon_P^{100}$ across all curves, is small, not exceeding one percentage point on average for any group.

Elasticity curve shape and the actual magnitude of expected streamflow change in response to a one percent change in precipitation do not necessarily correspond (Figure 3). For instance, in the summer, 78% of catchments exhibit a flat elasticity curve (Figure 2 summer: A; C). However, while skewed towards zero, the distribution of possible elasticity magnitude is widespread (Figure 3 summer: B), indicating that the streamflow response to a one percent change in precipitation in this group ranges from between about zero to two percent. Conversely, the distributions of magnitude for flat elasticity curves in winter
is concentrated around zero, indicating that streamflow across the majority of catchments has a very low responsiveness to





precipitation variation in this season. In other words, a flat elasticity curve indicates that low and high flows have approximately the same response to precipitation changes within a particular catchment, but that response is not necessarily small or consistent across catchments with the same elasticity curve shape. The highest actual elasticity values are predominantly in the eastern U.S. in all seasons. High magnitude elasticity values also occur in the Pacific Northwest particularly in the fall, winter, and summer seasons.

Panel regression models provide a robust estimation of the cluster-average points and allow for the calculation of confidence intervals around the points which comprise each $\varepsilon_{c,P}$. The point estimates, $\varepsilon_P^{g;q}$, are all significant at the 99.99% confidence level at least. The interactions are also significant at the 99.99% confidence level, except for annual streamflow above the 65th percentile, where all interactions are significant at the 95% confidence level, at least, except for the highest annual flow (100th percentile) for which the interaction is not significant. This means that the $\varepsilon_P^{g;q}$ estimates are statistically significantly different from one another for each of the clusters in every temporal scale and every percentile, with the exception of the highest annual streamflow. The actual magnitude of the elasticity estimates for the maximum annual streamflow is not statistically different across the groups (Figure 3 annual: B). Otherwise, the $\varepsilon_{c,P}$s resulting from the panel regression models are remarkably similar to those which we generate using single catchment linear models, lending credibility to the elasticity estimates.

## 3.1 Attribution and predictability of between-catchment variation in $\varepsilon_{c,P}$

We perform a multivariate variable importance analysis using random forest models to determine to what extent catchment attributes predict elasticity curve shape. The following catchment characteristics are included in this analysis: Aridity index, DFI, BFI, Slope of the flow duration curve (calculated at the 0-33rd, 33-66, and 67-100th percentiles), latitude, coefficient of variation for daily streamflow in each season, mean annual temperature, mean catchment elevation, drainage area, mean catchment slope, and snow fraction, as well as, precipitation and streamflow seasonality and timing metrics. Averaged over 10 iterations each, the random forest model accurately predicted class membership in approximately 70% of cases at the annual timescale, 95% for fall, 79% for winter, 63% for spring, and 79% for summer, all rounded to the nearest integer.

For each temporal scale, different variables were selected as the best predictors of cluster membership using both the Gini coefficient and the mean decrease accuracy metric. For both the annual and summer periods, fdc$_{bl}$ was the best predictor for every iteration of the random forest model. At the annual timescale, DFI, fdc$_b$ and aridity are the second and third best predictors of cluster membership depending on the model run. The second and third best predictors for summer class membership vary widely between iterations. In winter, the best predictors for both metrics were either average annual temperature, or the time delay between the least important precipitation season and low streamflow season. In addition to these metrics, mean catchment elevation and other seasonality metrics were frequently selected as the second or third most important predictors for winter depending on the model run. For fall, the time delay between the least important precipitation season and low streamflow season, mean catchment elevation, and BFI were the top three predictors in the majority of iterations of the model for both metrics and typically had very similar mean decrease accuracy scores and Gini coefficients. No variable was clearly the best predictor of cluster membership in springtime, as over the course of 10 model runs, eight different variables had the highest Gini coefficient or mean decrease accuracy score.





## 4   Discussion

In this paper, we use multivariate statistical models to investigate whether streamflow elasticity to precipitation change varies across the distribution of streamflow at the annual and seasonal timescales. We then use a clustering algorithm and random forest regression model to examine whether that variation is systematic and predictable.

By creating elasticity curves, which represent the range of elasticity across the streamflow distribution (Figure 1), we show that at the annual and seasonal timescales, the highest streamflow percentiles are typically more responsive to long-term precipitation change relative to lower streamflow percentiles in the same catchment and time period. This is especially true for annual elasticity and in the spring, winter, and fall. The finding that low flows are less responsive to precipitation change than higher flows is in line with existing theory. Low flows are typically sustained by groundwater, saturated soils, and surface water storage which require precipitation for recharge, but for which the effects of changes in precipitation are inherently delayed and moderated (Smakhtin, 2001; Price, 2011; Gnann et al., 2021).

There are, however, catchments which do not have positively sloping elasticity curves at some timescales. Approximately 7% of catchments at the annual timescale and 8% in summer are clustered into groups with generally negative trends, indicating that low flows are relatively more responsive to precipitation than are higher streamflow percentiles. Further, the elasticity curves of roughly 31% of catchments in fall, 78% in summer, 65% in spring, and 26% in winter are nearly flat, having very low slopes for the majority of the curve, with $\varepsilon_P^q$ estimates only increasing marginally for the highest streamflow percentiles.

The best predictors of elasticity curve shape are those related to the hydrologic storage capacity of the catchments. For instance, $fdc_{bl}$, the most important catchment attribute at the annual timescale and in summer, provides information about a catchment's ability to sustain flows of a certain magnitude during the dry season. The flow duration curve (fdc), here calculated using daily streamflow for the entire study period, is a cumulative frequency curve which shows the percentage of time that a certain magnitude of streamflow is equaled or exceeded (Searcy, 1959). When the slope of the fdc is steep, it indicates that a catchment has highly variable streamflow predominantly originating from direct runoff, and when the slope is relatively flat, it suggests the presence of surface or groundwater storage, which equalizes flow. At the low end of the fdc (here $fdc_{bl}$), a flat slope points to the presence of long-term storage within the catchment, while a steep slope indicates that very little exists (Searcy, 1959). Similarly, baseflow is the portion of streamflow that is derived from groundwater and other delayed sources (Smakhtin, 2001), and a low BFI indicates a catchment for which a majority of streamflow comes from direct runoff. We have defined two baseflow metrics, BFI and DFI, a delayed flow metric over a longer time span (Gnann et al., 2021; Stoelzle et al., 2020), both of which are frequently important predictors of elasticity curve shape. Further, while snow fraction was not necessarily the most important predictor in cold months, temperature, latitude, elevation, and the time gap between the most important precipitation and streamflow season, attributes which speak to precipitation type and snow dominance, were.

Storage components consist of anything ranging from surface waterbodies such as wetlands, to snow cover, and ground water influxes, all of which interact with fluvial systems on different timescales. Catchments with relatively flat elasticity curves in cold months (winter and fall), are typically those at high latitudes which receive higher percentages of precipitation as snow, or those in the semi-arid southwestern region which are predominantly fed by snow melt upstream (Li et al., 2017). These





curves are flat and have actual elasticity estimates which are heavily skewed towards zero (Figure 3 winter: A; B; fall: A;
B) because snow melt does not usually occur in winter or fall. However, at the annual timescale, the same catchments have
actual elasticity values ranging from less than 1 for low flows to around 2 for the highest annual flows because the streamflow
response is delayed, but occurs within the same year. In the fall, there are additionally catchments in Florida and scattered
along the southern coast with relatively flat elasticity curves, potentially due to increased storage within the catchment area
e.g. as wetlands.

Flat elasticity curves are present across the majority of the country during the summer (Figure 2 summer: A; B; C), indicating
that the response of streamflow to summer precipitation is similar across all flow percentiles in these catchments. Similar to
winter and fall, the flat elasticity curves tend to have higher BFI and DFI and lower $fdc_{bl}$ values than type A or C curves. Many
of these catchments have average actual elasticity values which approximate 0, indicating that in-season precipitation has little
to no influence on seasonal streamflow, however, others have larger average actual elasticity values, often greater than one
(Figure 3 summer: A; B), which in turn, implies summer precipitation has a substantial influence on summer streamflow, but
that the influence is consistent across the distribution. This differs from a majority of cases in other seasons and at the annual
timescale, for which the influence of precipitation on streamflow is magnified in higher streamflow percentiles.

It has been shown that high flow magnitudes are driven by the combined influence of precipitation events and antecedent
soil moisture (Ivancic and Shaw, 2015; Slater and Villarini, 2016). Summer is a period of relative soil moisture deficit (Koehn
et al., 2021) and high potential evaporation. It is plausible therefore that the non-zero magnitude, flat, elasticity curves in the
majority of the study region during this period are emblematic of the relationship between antecedent wetness, precipitation,
and streamflow. In other words, because of a soil moisture deficit, the precipitation changes are not typically magnified in
higher streamflow percentiles in the majority of catchments (78%) during this period. This does not, however, explain the
relative homogenization of the elasticity curve structure in the spring, a period in which soil moisture recharge is likely to
occur. Instead, it seems probable that the flatness of the elasticity curve shape, despite a persistently broad range of elasticity
magnitudes in spring (Figure 3 spring: B), may be due to the fact that streamflow is the least variable on average in springtime
compared to the other seasons, as determined by the coefficient of variation (CV) of the daily streamflow measurements, and
that springtime is the low flow season in only 24 catchments. In other words, the lowest flows in spring may be more heavily
driven by runoff from precipitation rather than storage as compared to other seasons. This hypothesis is further supported by the
cluster-specific CV distributions at other timescales – where type B elasticity curves correspond to catchments with relatively
low variability (Figure 4 spring). The shape may also reflect, in part, the climatic drivers dominant over different regions.

The range of type B elasticity curves which is present across the seasons is washed out at the annual scale, demonstrating that
the catchment storage which leads to a uniform response across the distribution of streamflow generally operates at a timescale
of less than a year (Figure 3). Type A elasticity curves with a strong signal exist across temporal scales, in catchments which
have relatively low BFI and DFI and steep middle sections of the flow duration curve, $fdc_{b}$, as compared to type B and weak
type A signals (Figure 4).

Interestingly, at the annual timescale, curve type C (negative) catchments are in some ways similar to those with strong curve
type A (positive) signals, in that they both have low snow fraction, low BFI, and steep $fdc_{b}$ slopes. They differ, however, in a





number of other attributes, most notably, DFI and slope of the low end of the flow duration curve, $\text{fdc}_{bl}$. This difference indicates

that while streamflow in catchments exhibiting both types of curves is predominantly rain-fed, those exhibiting strong type A curves are better able to sustain low flows as compared to type C catchments. Catchments with type C curves have very flashy low flow behavior. We controlled for ephemeral streams in this study in order to simplify our methodology, but including those catchments may increase the prevalence of type C curves. The elasticity curve for the individual sites, and the panel estimated Type C curves have wide confidence intervals, indicating lower robustness in the estimation of this group overall (Figure 2. The

strong type C cluster at the annual timescale (especially as estimated by the panel regression model) also exhibits a positive slope above the 35th percentile of streamflow. While speculative, these results suggest that type C curves may differ from positive $\varepsilon_{c,P}$s predominantly in that they exhibit highly flashy low flow behavior (Figure 4).

## 4.1 Limitations and context

### 4.1.1 Example catchments

Elasticity curves computed at individual sites typically have wide confidence intervals and should be applied cautiously, but we select three sites may serve as an example of the elasticity curve concept, and the limitations of the approach in context. The three catchments provide a detailed example of the approach and mechanistic insights. These examples coincide with Gnann et al. (2021) who proposed a framework for incorporating regional knowledge into large sample hydrology when studying baseflow processes and drivers. They include detailed examples of the processes controlling baseflow and delayed

flow partition in catchments in different regions of the U.S., some of which happened to be included in our analysis. The non-normalized elasticity curves for Turnback Creek above Greenfield (gage id: 06918460), Current River at Van Buren (gage id: 07067000), and Reddies River at North Wilkesboro (gage id: 02111500) are included in (Figure 5). As discussed in Gnann et al. (2021) Turnback Creek above Greenfield and Current River at Van Buren both lie within the Ozarks Plateau aquifer system, a karstic region with substantial baseflow components, while Reddies River at North Wilkesboro is located in the Appalachian

mountains in North Carolina. Despite being located near one another, 07067000 lies over the Ozark aquifer, a more mature karstic environment, with comparatively more long-term storage and higher DFI (0.4) and BFI (0.7) as compared to 06918460 (DFI: 0.1; BFI: 0.5) (Gnann et al., 2021). Conversely, 02111500 is physically distant from the other two catchments and has a different geological profile (Zimmer and Gannon, 2018), but, has substantial seasonal and stable storage components resulting in high DFI (0.4) and BFI (0.7) values compared to both of the Ozarks catchments. Catchment attributes for each of these sites

are presented in Table 1.

At the seasonal timescale, both of the Ozarks catchments (Figure 5; in purple) are consistently classified as the same curve type. However, several things are apparent: first, in non-normalized format, as presented in Figure 5 A, it is clear that the catchment with young Karstic geology (06918460) and comparatively less long-term storage experiences a higher absolute magnitude of elasticity to precipitation (Figure 5 A) compared to its counterpart. This is particularly clear in summer, where

the curve shape is similar (Figure 5 B) but the estimated magnitude of elasticity differs by more than one percentage point. Second, despite having relatively similar curves at the seasonal timescale, these two catchments exhibit different behavior at



the annual timescale, where 06918460 has a strongly positive signal and 07067000 has a weakly positive signal, demonstrating the association between increased long-term storage and a less steeply sloping elasticity curve. At the annual timescale, the elasticity curves of these two catchments demonstrate the nuance required in interpreting the classification system – both curves

span a similar total range of elasticity, however, the overall condition of the strongly positive curve (06918460) is steeper, as a large portion of the increase in the elasticity curve for 07067000 occurs between the 95th and 100th flow percentiles. Further, the more physically distant catchment (02111500; Figure 5, represented in green), has relatively similar characteristics to 07067000 (Table 1) and exhibits similar curve structure at the annual and seasonal timescales, although with a slightly flatter overall condition.

Informative in the aggregate, the elasticity curve concept is limited in several ways, some of which are apparent in these examples. First, while curve shape is approximately consistent within the clusters, there is a margin of error around the groupings. The choice of the number of clusters per temporal scale was carefully considered in the interest of parsimony, so some catchments inevitably exhibit behavior outside of the norm. Further, the shapes of the curves are not always smooth, as is evident in the example catchment 06918460, where a substantial decrease in elasticity is evident between the 80th and 95th

percentiles at the annual timescale. The intention of this paper is to provide an introduction to the concept in a large-sample context and additional research is needed to determine the extent to which minor variations in shape may be due to statistical noise or physical processes. Thus the suitability of the concept for application to small scales remains to be established.

The lm-constructed curves or point estimates in individual catchments may deviate substantially from the cluster average, may be comprised of insignificant point estimates, or may violate assumptions of the regression approach used. For instance,

depending on the streamflow percentile, the residuals of between 68 ($\varepsilon_p^0$) and 78 ($\varepsilon_p^{100}$) percent of the single catchment lms were normally distributed as estimated by a Shapiro-Wilks test with an alpha level of 0.01, and between 75 ($\varepsilon_p^0$) and 80 ($\varepsilon_p^{100}$) percent had a Durbin-Watson test statistic of greater than one, indicating that autocorrelation was not a serious concern in these sites. This means that the normality assumption was violated in around 20 to 30% of catchments and the non-autocorrelation assumption was violated in 20-25% of catchments. The fixed effects panel regression approach helps to mitigate these concerns,

lending credibility to the aggregated curves, but the reader is cautioned that application at the scale of a single catchment may carry substantial uncertainty. Further, the single catchment multivariate regression approach which we have taken here is a standard method for calculating point estimates of elasticity, however, this approach does not accommodate the possibility of non-linear elasticity, e.g. the possibility that a one percent and a 10 percent difference in precipitation are not linearly related. This work only considers the elasticity of streamflow magnitude, a singular component of streamflow which may not fully

capture the influence of precipitation variability. Finally, the selected clusters depict whether curves are generally increasing or decreasing but do not take into account the exact shape of the curves themselves, for instance, at which percentiles the slope begins to increase or decrease. In some instances, the curves for individual sites do not follow the precise curve types for which we have named the clusters. For instance, while the average curve in a cluster may be type A: strong, an individual curve may be type A: weak, etc. For this reason, we have presented the single catchment data with the interquartile ranges of curve

estimates, and recommend caution when estimating elasticity curves or even elasticity magnitude for individual locations.





The work presented in this manuscript represents an introduction to elasticity curves. This concept may support understanding of how changes in water storage effects streamflow response over time, building on work such as Saft et al. (2015) and Saft et al. (2016), and providing insight into the implications of climate change for the hydrological cycle and the rainfall runoff relationship. Further, we have presented panel regression models as a tool for more robust elasticity estimation – a method which may be well suited to regional calculation of elasticity and estimation in ungauged sites.

## 5 Conclusions

In this paper, we introduce a new concept for understanding and classifying streamflow response to precipitation. Representing streamflow elasticity to precipitation as a curve which reflects the range of responses across the distribution of stream flow within a given time period, instead of a single point estimate, provides a novel lens through which we can interpret hydrological behavior. We have shown that $\varepsilon_P$ estimated from the central summary of streamflow, e.g. the annual median, does not provide a complete picture of streamflow change. We have demonstrated that elasticity curve shape, i.e. the response of different flow percentiles relative to one another, can be understood separately from between-catchment variation in the actual magnitude of streamflow elasticity associated with a one percent change in precipitation.

We identify 3 typical elasticity curve shapes:

Type A – in which low flows are the least and high flows are the most responsive. The majority of catchments at the annual, winter, and fall timescales exhibit this behavior.

Type B – in which the response is relatively consistent across the flow distribution. At the seasonal timescale, many catchments experience a consistent level of response across the flow regime. This is especially true in snow-fed catchments during cold months, when the actual elasticity skews towards zero for all flow percentiles while precipitation is held in storage. Consistent response is seen across the majority of the country during spring when streamflow is comparatively stable and in summer when evaporation demand is high and soil moisture is low.

Type C – where low flows are the most responsive to precipitation change. These catchments are dominated by highly flashy low flow behavior.

Depending on the timescale examined, we predict elasticity curve type with fairly high accuracy, ranging from 95% in the fall to 63% in the spring, using catchment and hydrologic characteristics. The best predictors of curve type include the low end of the slope of the flow duration curve, mean annual temperature, seasonality, mean catchment elevation, and the baseflow index. All of these attributes relate to hydrological storage and release timing.

## 6 FIGURES





**Table 1.** Attributes of example catchments: Turnback Creek above Greenfield (gage id: 06918460), Current River at Van Buren (gage id: 07067000), and Reddies River at North Wilkesboro (gage id: 02111500). BFI is the baseflow index, DFI is the delayed flow index, the slope of the flow duration curve between the 33rd and 66th percentile is $fdc_b$, between the 67th and 100th percentile (low flows) is $fdc_{bl}$, and between the 1st and 32nd (high flows) is $fdc_{bu}$. Average annual T is the annual mean daily temperature, Annual PET and Annual P are the average annual total potential evaporation and precipitation, LAT is the latitude, SF is the snow fraction, Max and Min P season are the most and least important precipitation seasons respectively, and Max and Min Q season are the most and least important flow seasons respectively.

| STAID | BFI | DFI | $fdc_b$ | $fdc_{bl}$ | $fdc_{bu}$ | Average annual T (C) | Annual PET (mm) | Annual P (mm) | RC | Aridity | Drainage area $(km^2)$ | LAT | SF | Max P season | Min P season | Max Q season | Min Q season |
|---|---|---|---|---|---|---|---|---|---|---|---|---|---|---|---|---|---|
| 02111500 | 0.7 | 0.4 | 1.6 | 5.5 | 10.9 | 12.8 | 774.2 | 1335.8 | 0.4 | 0.6 | 233.7 | 36.2 | 0 | summer | fall | spring | fall |
| 06918460 | 0.5 | 0.1 | 3.5 | 8.3 | 14.4 | 13.4 | 843 | 1159.5 | 0.3 | 0.7 | 650.7 | 37.4 | 0 | spring | winter | spring | fall |
| 07067000 | 0.7 | 0.4 | 1.7 | 2.5 | 13.1 | 13.2 | 826.2 | 1183 | 0.4 | 0.7 | 4349 | 37 | 0 | spring | fall | spring | fall |



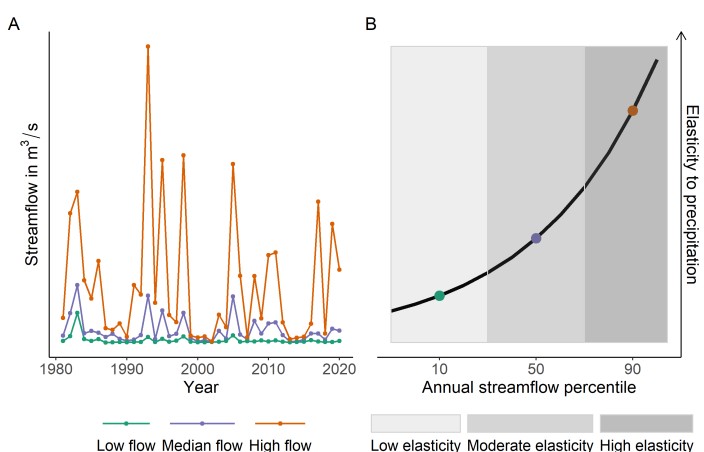

**Figure 1.** Conceptual diagram demonstrating how to read an elasticity curve, where plot panel A. Shows hypothetical high, low, and median annual streamflow ($10^{th}$, $50^{th}$ and $90^{th}$ percentiles of the flow distribution in each year ) and plot panel B. Shows the hypothesized relative elasticity of each of these streamflow percentiles to changes in annual preciptiation. For simplicity, this diagram show only 3 points, but a typical curve in this study would normally include 21 points.



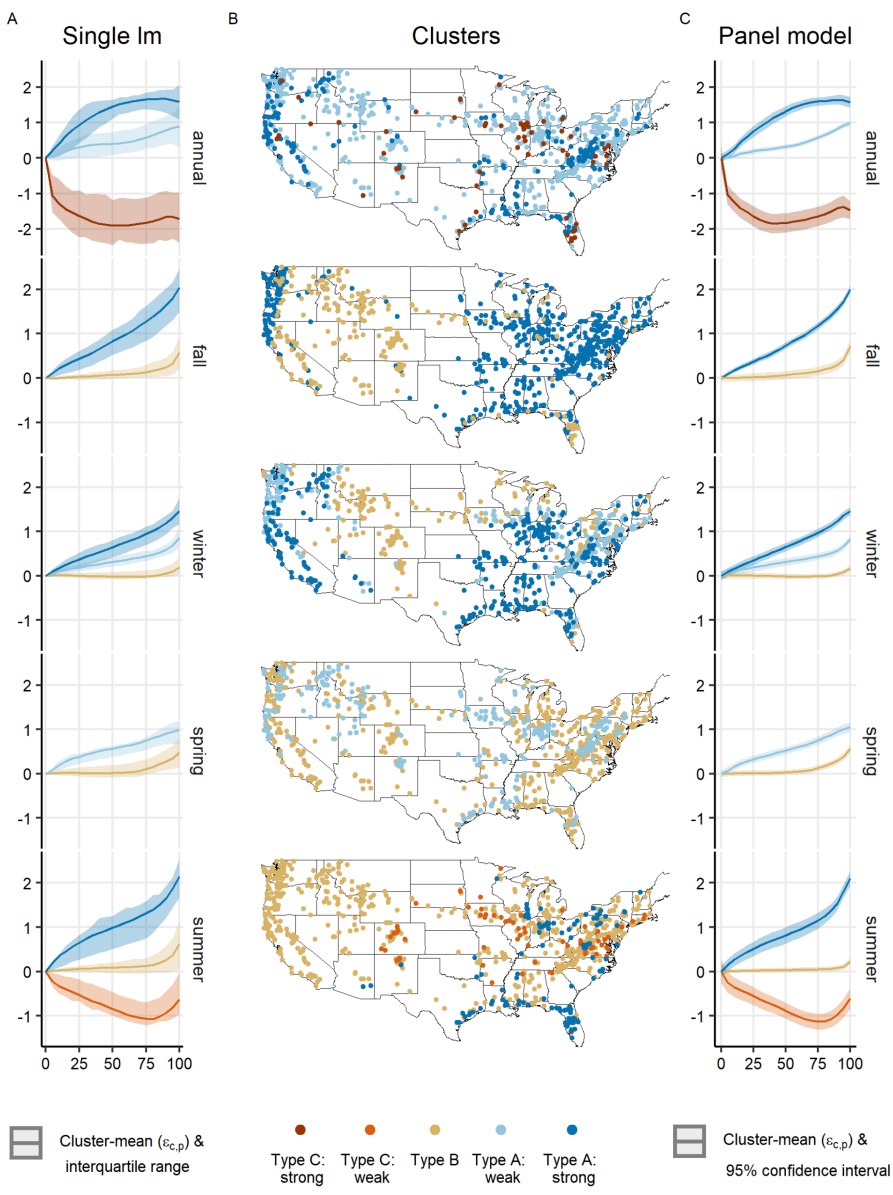

**Figure 2.** Normalized elasticity curves. Panel A shows the curves resulting from the single catchment linear models (lm) where the line is the mean of the distribution of elasticity point estimates (for a cluster of sites) and the bands are the inter-quartile range. Panel B shows the geographic distribution of cluster membership and panel C shows the elasticity curves as estimated from the causal panel model, where the line is the estimated normalized curve and the bands are the normalized 95 percent confidence interval. Because the curves and confidence intervals have been normalized, a confidence interval which overlaps with zero does not imply that the estimate is not statistically significant. Note the Y-axis scales differ between the annual and seasonal plots, and the Y-axis is shared in panels A and C. Spring and fall have 2 clusters while winter, summer, and annual have 3. The x-axis shows streamflow percentiles (0-100) and the y-axis show normalized elasticity.



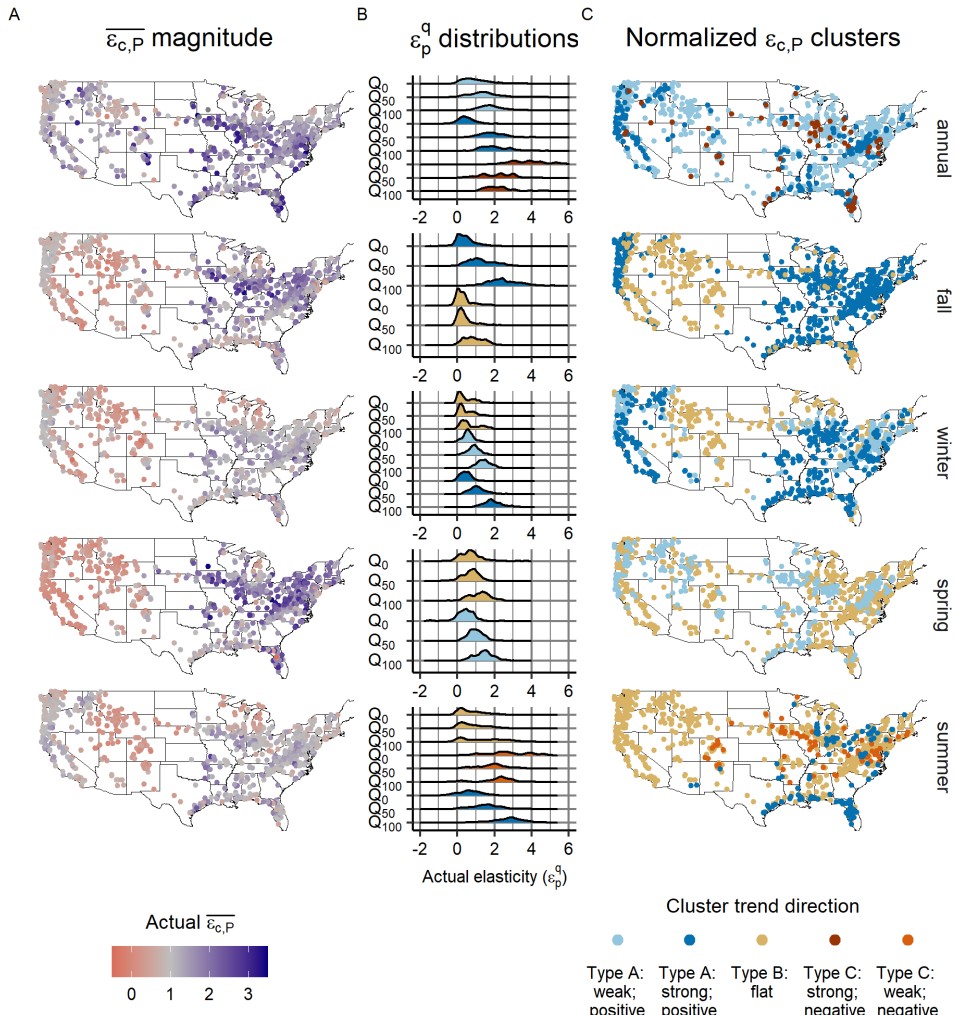

**Figure 3.** Actual elasticity compared to normalized elasticity curves. Panel A shows the geographic distribution of the means of the non-normalized, site specific, elasticity curves. These values are often referred to as the actual elasticity in the text. Smaller mean elasticity values (less responsive) are in lighter purple and higher mean elasticity values are in darker purple. Panel B shows the distributions for non-normalized point estimates of elasticity at the lowest, median, and highest streamflows (Q0, Q50, Q100) in each timescale (annual, winter, spring, summer, fall). The distributions in Panel B are colored according to the cluster membership of the normalized curves, the geographic distribution of which is shown in Panel C.

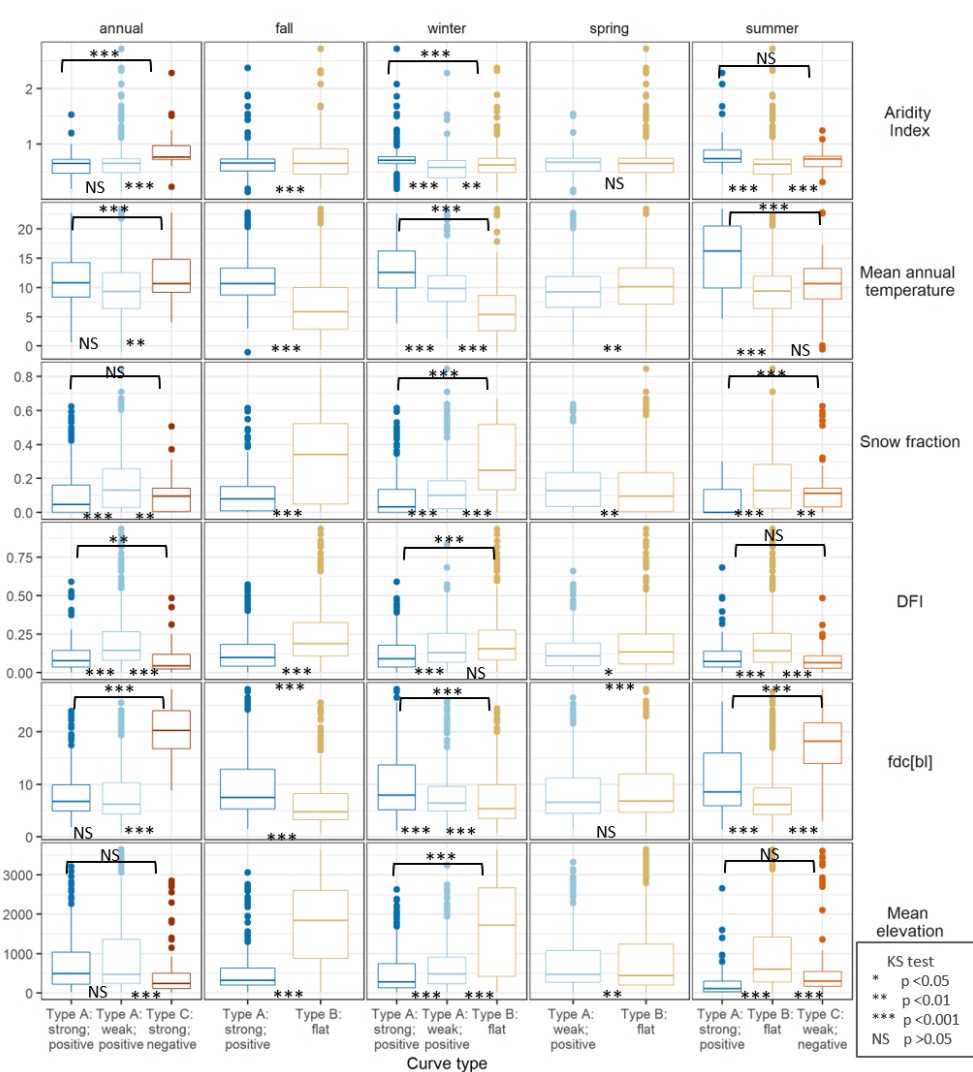

**Figure 4.** Boxplots showing the distributions of static catchment attributes for the annual and seasonal scales (columns) and by cluster membership (boxplot color). Asterisks at the bottom of each panel indicate if the distribution of neighboring boxplots is significantly different, using the Kolmogorov-Smirnov (KS) test. Asterisks at the top of each panel indicate if the distributions of the first and last boxplots are significantly different. Boxplots are included only for attributes (rows) which are important in the RF analysis, and which can be represented by continuous numeric values, so seasonality metrics are excluded here.



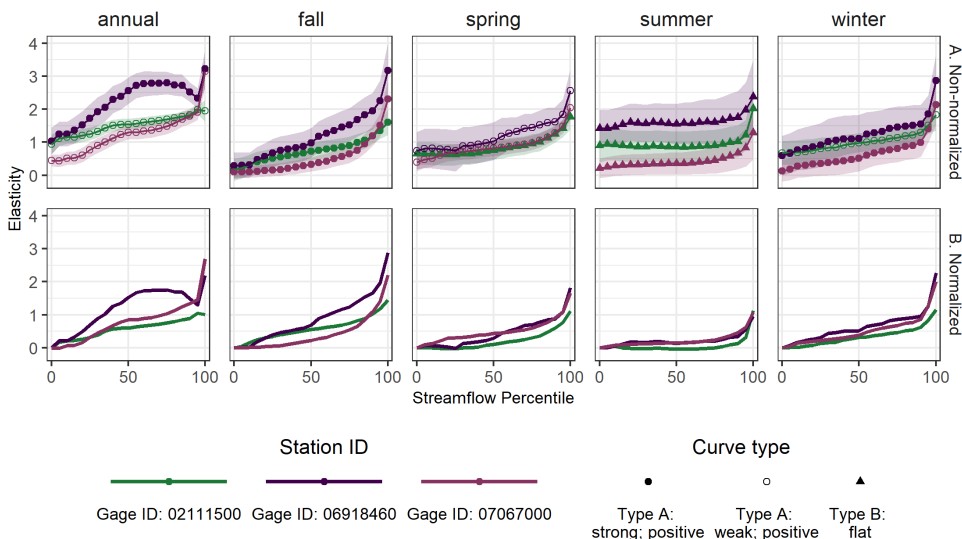

**Figure 5.** Examples of elasticity curves from three catchments: Turnback Creek above Greenfield (gage id: 06918460), Current River at Van Buren (gage id: 07067000), and Reddies River at North Wilkesboro (gage id: 02111500). Panel A. shows the non-normalized curves to demonstrate actual elasticity, and panel B. shows the normalized curves to demonstrate the similarity in curve form. Catchments located near one another geographically are both represented in shades of purple. Point shape represents the curve type and the ribbon represents the 95 percent confidence interval. Points and confidence intervals have been removed from panel B to improve visibility, but the curve types and confidence bands are consistent across both panels.





# 7 Appendix

**Table A1.** Description of catchment attributes considered in the explanatory analysis.

| Variable | Method | Description |
|---|---|---|
| DFI | Smoothed minima method, 90 day | Calculated using the R package delayed flow (Stoelzle, 2022) |
| BFI | Smoothed minima method, 5 day | Calculated using the R package delayed flow (Stoelzle, 2022) |
| Snow fraction | | Proportion of precipitation falling in months when the average temperature is below 0 |
| Permeability | | Average catchment permeability (mm/hr) (Falcone, 2017) . |
| Aridity index | Aridity$=(\overline{(\overline{PET}/\overline{P})})*100$ | |
| Runoff Coefficient | RC$=((\overline{Q}/D)/\overline{P})*100$ | |
| fdc$_b$ | fdc$_b = \frac{\ln(Q_{33})-\ln(Q_{66})}{(0.66-0.33)}$ | Slope of the annual flow duration curve calculated with daily flow between the 33rd and 66th flow exceedance probabilities |
| fdc$_{bu}$ | fdc$_{bu} = \frac{\ln(Q_0-Q_{32})}{0.32}$ | Slope of the annual flow duration curve calculated with daily flow between the 0th and 32nd flow exceedance probabilities |
| fdc$_{bl}$ | fdc$_{bl} = \frac{\ln(Q_{67})-\ln(Q_{100})}{(1-0.67)}$ | Slope of the annual flow duration curve calculated with daily flow between the 67th and 100th flow exceedance probabilities |
| Annual temperature | | Mean annual temperature |
| Mean catchment elevation | | In meters (Falcone, 2017) |
| Latitude | | Latitude at gage site (Falcone, 2017) |
| Drainage area | | In $km^2$ (Falcone, 2017) |
| Average catchment slope | | In Degrees (Falcone, 2017) |
| Coefficient of variation | CV$=$sd$(Q)/\overline{Q}$ | Calculated in each time step using daily streamflow |



*Code and data availability.* Datasets are publically available as of December 2, 2022 at: total upstream dam storage, average annual runoff, drainage area, mean catchment elevation, latitude and average catchment slope are available through GAGES II at (https://water.usgs.gov/GIS/dsdl/basincha and in Falcone (2011); watershed boundaries are also available through GAGES II at (https://water.usgs.gov/GIS/dsdl/gagesII_9322_point_shapefile.zip) and in Falcone (2017); climate data is available from PRISM at (https://www.prism.oregonstate.edu/recent/) and can be downloaded using the prism R package (Edmund and Bell, 2020). Streamflow data

can be downloaded from the National Water Information System (NWIS) using the R package dataRetrieval (Cicco et al., 2022). R code for the complete analysis is available as of December 2, 2022 at: https://doi.org/10.5281/zenodo.7391227 (Anderson, 2022)

*Author contributions.* Conceptualizaton, Methodology: BJA, MIB, and LJS; Data curation, Formal analysis, Investigation, Project administration, Software, Visualization, Writing – original draft preparation: BJA; Supervision, Validation: LJS, SJD, MIB; Writing – review & editing: LJS, SJD, MIB, BJA

*Competing interests.* At least one of the co-authors is on the editorial board of HESS. The authors have no other competing interests to declare.

*Acknowledgements.* LJS is supported by UKRI (MR/V022008/1). BJA is supported by the Clarendon scholarship and Hertford College, Oxford. We additionally thank the attendees of the 2022 STAHY workshop, the hydrology research group at the University of Freiburg, and the Water lab at the University of Oxford for their valuable input. In particular, we thank Richard Vogel, Michael Stoelzle, Markus Weiler,
and Kerstin Stahl whose comments led to methodological and conceptual improvements in the work.



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
