# Peer review of "Elasticity curves describe streamflow sensitivity to precipitation across the entire flow distribution"

_Hydrology and Earth System Sciences, 2022_

## Referee Comment (RC2)

Review comments on "Elasticity curves describe streamflow sensitivity to precipitation across the entire flow distribution" by Anderson et al.

**General Comments**

This manuscript proposes a new concept in which streamflow elasticity is estimated across the full range of streamflow percentiles in a large-sample context, which is called "elasticity curve" by authors. The aim is to develop a more complete depiction of how streamflow responds to precipitation. They find three different elasticity curve types which characterize this relationship at the annual and seasonal timescales in the USA, based on two statistical modelling approaches, a panel regression which facilitates causal inference and a single catchment model which allows for consideration of static attributes. The idea is novelty and fits well with aims and objectives of HESS. This was why I accepted the review invitation. However, there are significant shortcomings in current version so that I have to recommend a rejection (below specific comments for detail).

**Specific Comments**

First of all, authors clearly misunderstand the concept of elasticity precipitation of streamflow proposed by Schaake (1990) and Sankarasubramanian et al. (2001). The original formula is as:

$$\epsilon_{p}(P,Q) = \frac{dQ/Q}{dP/P} = \frac{dQ}{dP}\frac{P}{Q}$$
(1)

However, the difficulty with this elasticity is that we never really know dQ/dP, which is often estimated from a hydrological model and, of course, the form of the hydrological model is always unknown and validation of such a model remains a fundamental challenge (Sankarasubramanian et al. 2001; Fu et al., 2007).

In order to solve this problem, Sankarasubramanian et al. [2001] introduced a specific case of (1) at the mean value of the climatic variable:

$$\epsilon_p(\mu_p, \mu_Q) = \frac{dQ}{dP}|_{P=\mu_P} \frac{\mu_P}{\mu_Q}$$
(2)

They (Sankarasubramanian et al. 2001 further verified that the non-parametric estimator:

$$e_{P} = median\left(\frac{Q_{t} - \overline{Q}\,\overline{P}}{P_{t} - \overline{P}\,\overline{\overline{Q}}}\right) \tag{3}$$

is a robust estimator of the precipitation elasticity of streamflow for a wide class of hydrological models that does not depend on the form of the hydrological model.

This is the formula that has been wildly used in the literature to estimate the precipitation elasticity of streamflow. That is to say, the elasticity is the median value of ratio of annual streamflow anomaly in terms of long-term means to precipitation anomaly, not the long-term mean for the 50th percentile of streamflow as author claimed.

I do understand that there are some exceptions in the literature not to take this median value. For example, the two-parameter elasticity to include temperature (Fu et al., 2007) is to plot every annual ratio is plotted in a 2-d space or fitted a linear regression with these two anomalies (Zheng et al., 2009).

$$e_{P,\delta T} = \left(\frac{Q_{P,\delta T} - \overline{Q}\,\overline{P}}{P_{P,\delta T} - \overline{P}\,\overline{\overline{Q}}}\right) \tag{4}$$

$$\Delta Q_i / \overline{Q} = \epsilon \cdot \Delta X_i / \overline{X}. \tag{8}$$

My main scientific concern is Eq 2 of the manuscript, which is the base of this study. This does not make any scientific sense, because the same percentile of streamflow and precipitation could happen in different time of year. For example, 95th percentile of streamflow is located in June and 95th percentiles of precipitation/PET could be in December. How possible to build a regression model between them?

$$\ln(Q_{i,t}^q) = \alpha_{i,t} + \varepsilon_P^q \ln(P_{i,t}) + \varepsilon_E^q \ln(E_{i,t}) + \eta_{i,t}^q$$

In addition, this approach requires non-zero daily streamflow for the entire study period, i.e., it cannot be applied to ephemeral rivers and streams, which limits its applications. I am surprised that it includes some rivers in Nevada and Arizona states where the number of rainfall days in a year is only 30-60 days. How can it result in a non-zero streamflow days?

**Reference**

Fu, G., S.P. Charles, F.H.S. Chiew (2007), A two-parameter climate elasticity of streamflow index to assess climate change effects on annual streamflow, Water Resour. Res., 43 (2007), p. W11419, 10.1029/2007WR00589

Sankarasubramanian, A., R. M. Vogel, and J. F. Limbrunner (2001), Climate elasticity of streamflow in the United States, Water Resour. Res., 37, 1771–1781.

Schaake, J. C. (1990), From climate to flow, in Climate Change and U.S. Water Resources, edited by P. E. Waggoner, chap. 8, pp. 177–206, John Wiley, New York.

Zheng, H., L. Zhang, R. Zhu, C. Liu, Y. Sato, Y. Fukushima (2009), Responses of streamflow to climate and land surface change in the headwaters of the Yellow River Basin. *Water Resour. Res.* 45, W00A19.

---

## Author Comment (AC1)

Comments/Text of Keirnan Fowler (Reviewer 1) posted in black, our text in blue with additions to existing text in red.

This paper examines the common concept of streamflow elasticity and takes it one step further, examining the sensitivity of different flow percentiles to changes in precipitation. This means the impact of climate changes can be examined separately for high flows and low flows. I find this to be a worthwhile extension to an existing widely-used method. It allows the elasticity concept to be more closely related to problems of societal interest such as ecological sensitivity to changes in low flows and impacts on infrastructure due to changes in high flows.

I find the manuscript to be close to publication standard already. The methods used are rigourous, the writing is usually quite clear, the findings are well supported by quality figures, and the paper is relatively complete. I offer the following comments, in the hope of improving the paper from its already high standard. Points 1 and 7 are editorial (and thus subjective) and are suggestions only.

We first thank the reviewer for their comments and suggestions which are thorough and contribute positively to the development of the manuscript. We propose making the following 3 substantial revisions to the paper.

1. Primarily, we will move the majority of the discussion and results of the panel regression models to the appendix.
2. Second, we will re-work the abstract so that it more accurately reflects the paper's objectives and conclusions and will further clarify the aims and justification in the introduction.
3. Finally, we will adjust the phrasing around the methodology, particularly equation 2, and add additional examples in order to limit confusion about the approach.

Comments are addressed in more detail below.

1. OVERALL FRAMING OF PAPER. I think the abstract and introduction could be improved to frame the paper better and increase its impact. To me, the paper should primarily aim to be an introduction to a new concept (or, more precisely, a new variant on an existing concept), as per the existing line 400: "The intention of this paper is to provide an introduction to the concept [of elasticity curves] in a large-sample context". If this is indeed the goal, then the authors ought to aim to clearly establish: (1) the importance of the existing method; and (2) the need for the new method, couched in terms of the limitations of the existing method. Neither of these aims are achieved very well in the existing abstract and introduction.

Specifically, the tone of the introduction seems to take for granted that the existing method is important; it does not clearly explain its significance or what questions can be/have been answered by the method in the past. Likewise, although the introduction does go some way to answering (2) (line 62-63), it waits too long to do so and does not go into sufficient detail (saying only "abnormally high and low flows are associated with the greatest strain on hydrological systems"). Can we get a lot more detail here? Eg. for high flows, it could acknowledge/discuss that infrastructure is often designed according to estimates of flooding potential, so any changes to this potential are very important; likewise low flows are important eg. for riverine ecology among other things. Articulating these factors will help the reader understand why the new method is important, which will motivate them to keep reading. In my view, it is crucial to do this early, before they lose interest.

As for the abstract, the majority of the text is spent trying to articulate the different catchment "types" that have been defined for the example application. This would be fine if the paper was about a new system for classifying catchments. But if the paper is about introducing the concept of elasticity curves, then this detail is unexpected and unhelpful in the abstract. The abstract needs to be about the method, not this particular application. Readers can read the full paper if they want this sort of detail. My suggestion would be to focus the abstract on the importance/significance of the method and what it adds; limit the results to a handful (say two or three) things that were learned in the specific application. Note, existing text in the conclusion section, Line 427 - 434, contains some of the above elements and could be adapted for the abstract.

The objective of this paper is to both establish the concept of an elasticity curve, and to demonstrate how, in combination, the multiple elasticity estimates are informative separately from point estimates. We agree with the reviewer's assertion that this could be clearer in the introduction and abstract and that the abstract is overly focussed on the classification scheme. We propose adding additional clarifying sentences throughout these sections which clarify the aim and the relevance of the method. In addition, we propose re-writing the abstract so that it reads as follows:

"Streamflow elasticity is a simple approximation of how responsive a river is to precipitation. It is a ratio of the expected percentage change in streamflow for a 1% change in precipitation. Typically estimated for the annual average streamflow, we here propose a new concept in which streamflow elasticity  is estimated for multiple percentiles across the full range of the streamflow distribution in a large-sample context. This "elasticity curve" can then be used to develop a more complete depiction of how streamflow responds to climate. Representing elasticity as a curve which reflects the range of responses across the distribution of streamflow within a given time period, instead of as a single point estimate, provides a novel lens through which we can interpret hydrological behavior. As an example application, we calculate elasticity curves for 805 catchments in the United States and then cluster them according to their shape. This results in three distinct elasticity curve types which characterize the streamflow-precipitation relationship at the annual and seasonal timescales. Through this, we demonstrate that elasticity estimated from the central summary of streamflow, e.g. the annual median, does not provide a complete picture of streamflow sensitivity. Further, we demonstrate that elasticity curve shape, i.e. the response of different flow percentiles relative to one another, can be interpreted separately from between-catchment variation in the magnitude of streamflow change associated with a one percent change in precipitation. Finally, we find that available water storage is likely the key control which determines curve shape."

2. CLARITY OF METHOD. I feel there is a strong possibility of readers misunderstanding the method. Specifically, the focus on different flow percentiles (or ranges of percentiles) may lead readers to believe that the method only focusses on precipitation that falls during the relevant percentile/range. For example, the reader might believe that the method is asking "how sensitive is low flow to precipitation that falls concurrently with times of low flow?" whereas my understanding is that the intent is to use the same seasonal or annual average of precip & PET regardless of which flow percentile is in view. Is this correct? Can the authors make this clearer please? Perhaps via some more concrete examples? An explanatory figure may also help.

The reviewer has correctly understood what we have done, and we agree that this could be made clearer throughout the document with a series of examples. In addition to some changes in phrasing, we propose adding the following clarifying statement: "As presented in this study, the elasticity curve characterizes the sensitivity of different percentiles of annual and seasonal streamflow to changes in

the average annual or seasonal precipitation. For example, an elasticity of 0.5 for the 15[th] percentile of annual streamflow would indicate that a 1% change in the overall mean annual precipitation would correspond to a 0.5% change in the 15[th] percentile of annual flow."

Additionally, we will change the text which describes the model (currently lines 127-130) to read:

"where $Q_{i,t}^{q}$ is the natural logarithm of a streamflow percentile (q) calculated for time period (t) for catchment (i), $\alpha_{i,t}$ is the intercept, $\ln(P_{i,t})$ is the logarithm of catchment averaged mean daily precipitation for the time period of interest (year or season), and $\ln(E_{i,t})$ is the logarithm of catchment averaged mean daily potential evaporation in that period. Note that mean seasonal and annual climate time series are used, not percentiles equivalent to the streamflow percentile of interest (denoted with the superscript "q"). The point estimate of precipitation elasticity is represented by the regression coefficient: $\varepsilon_{P}^{q}$ and potential evaporation elasticity is represented by $\varepsilon_{E}^{q}$ The error term is $\eta_{i,t}^{q}$."

3. GREATER JUSTIFICATION OF "CAUSAL" NATURE OF ANALYSIS. The panel regression model is described as a "causal" model (eg. line 152). Can the authors please provide more justification for this? I am not an expert in this area, so I am looking for more information here - it seems to me as if this method is a variant on linear regression, with additional care to hold confounding factors constant. However, even if the authors manage to hold every available confounding factor constant, it does not resolve the problem that correlation does not imply causation. Have other authors made similar claims of panel regression, and what is their reasoning? Given there exist specialised causation methods (ie. methods that were directly formulated to try to distinguish correlation and causation such as https://doi.org/10.1126/science.1227079), it is not a claim that I would be making lightly. Even if the authors agree with me, this is no reason to change to experimental design; merely the way it is described.

Panel models have been widely used for causal inference, especially in combination with graphical models, like a directed acyclic graph (D.A.G), to make the modelling assumptions clear. We used in the creation of the models but did not include them with the manuscript. Examples of such studies include Anderson et al., 2022; Blum et al., 2020; and Yang et al., 2021 in hydrology and Ferraro et al., 2019 in environmental studies, as well as a wide range of other research in fields including econometrics and health sciences. In fact, even the simple linear regression model can be used for "causal inference" if the assumptions are explicit enough (Pearl, 2009) although this is not a particularly robust tool. Causal inference implies a different way of approaching a problem, where the intent is explicitly to infer relationships which persist across changing conditions and which cannot be clearly defined by distribution functions alone, relative to more typical statistical inference approaches which focus on assessing the parameters of a distribution and establishing statistical significance. There is an enormous amount of overlap in methodologies, but the underlying principles differ.

Two-way linear fixed effects panel regression models are robust to many aspects which would normally bias regression approaches and this makes them a useful tool for this purpose. For instance, many catchment-level confounding variables (which are time-invariant) are controlled for by the streamgage-specific intercepts as in (Blum et al., 2020). Additionally, they address the majority of omitted variable bias by requiring that confounding variables either be directly measured or be

invariant along at least one dimension of the data, for instance, time (Anderson et al., 2022; Nichols, 2007). Simple linear regression models, or any single-site regression models for that matter, cannot address this. We direct the reviewer to Section 4.3. in Anderson et al., 2022, for a more detailed description.

That said, we also recognize that "causal inference" is relatively new in hydrology and that the use of causal terminology in this way may be misunderstood by unfamiliar readers. This is especially true in the context of the current manuscript, where we have used a very simple model design, have not presented the D.A.G as part of the manuscript, and where the inclusion of causal language is not essential to the argument. Further, we agree that while panel regression models are a robust statistical tool which offer a range of benefits for hydrological science, their validity for this purpose relies on having well-constructed, explicit, modelling assumptions (Imai & Kim, 2021).

Thus, while we feel confident that our modelling assumptions are valid and are comfortable making causal inferences, we propose limiting causal terminology in the paper and will instead rely predominantly on language appropriate for statistical inference approaches (e.g. associations, etc.). We will additionally move the majority of the discussion of the panel models to the appendix, as is suggested by the reviewer in a comment below. We will retain mention of the panel regression model in the text to demonstrate the robustness of the approach presented in the paper and will briefly elaborate on its causal applications in the text when directing the reader to the appendix. Because of the panel regression model's robustness to omitted variable bias, we feel that it is important that it not be entirely excluded from the paper.

4. JUSTIFICATION OF TWO METHODS. The results of the two methods are very close (a key difference is the uncertainty bounds, but these may be closer than they look - see following point). The results are so close that one wonders whether the two methods are actually doing almost the same thing. Two recommendations arise from this:

   - Are both methods really needed? They are taking up valuable real-estate and they really detract from the story because it becomes more about comparing the methods rather than reflecting on what the results actually mean in this first-of-its-kind study. Perhaps one of the methods could be moved to an appendix?

   - If the authors elect to keep both methods, I suggest they bolster their justification for why the two methods are different.

The panel regression models lend credibility to the concept because they are substantially more robust than blending together individual single site models. Therefore, we will not exclude the approach from the paper. Instead, we will move the majority of discussion of the panel regression model design, results, and associated figures to the appendix. In the text, we will state that panel regression models with a similar parameterization as the single site models were applied and that the results were remarkably similar; in this way we can reduce the space taken up by model inter-comparison and increase discussion of the results.

5. REPORTING OF UNCERTAINTY. If the authors elect to retain both methods, then the following becomes relevant. With respect to Figure 2: At first glance, the uncertainty bounds appear to be the

key difference between the two methods. However, the comparison is apples with oranges, and if the authors correct this then the uncertainty bounds may be much more similar. Specifically, for the panel model, the confidence intervals are plotted, whereas for the single catchment models, IQR is plotted, which is a lot closer to prediction intervals (which plot uncertainty about individual predictions) than it is to confidence intervals (which plot uncertainty about the underlying assumed relationship). Is it possible to retain the IQR for the single catchment models and then swap to prediction intervals for the panel regression?

We thank the reviewer for this useful comment. We will calculate prediction intervals for the panel regression models, which can be included in the appendix with those figures.

6. LIMITATIONS. Regarding line 320-322, this speaks to a significant limitation here: the method assumes a flow response in the same time period. So it might be that certain flows are very sensitive to changes in climate, but if this sensitivity is subject to a delay longer than the time period used, this will not be detected here. Is this true? If so, I suggest this be included / discussed.

This is true, but we do not feel that it is detrimental to the work. This is a typical assumption of hydrologic elasticity work, although it is rarely discussed as explicitly as we have done, and while this is particularly relevant at the seasonal scale, at the annual time scale, this seasonal element is less influential.

Basically, an elasticity estimate is always just going to provide an approximation of how responsive streamflow is to precipitation change in the time period you are looking at, which is why storage and evaporation are such important controls on elasticity. For example, if storage is seasonal in nature (e.g. snow), seasonal streamflow elasticity (e.g. in winter) might be close to 0, but the elasticity of summer streamflow to winter precipitation might be quite high, because local hydrology relies on winter snow melt to drive dry season flows. The method doesn't assume a flow response in the same time period, instead, the method sheds light on the extent to which a response occurs in that time period without ruling out the possibility of a delayed response. Specific investigation into delayed responses would be an interesting direction for future work, but is not the point of this paper. We propose adding the following clarifying comment at the end of the paragraph referenced, and may add further clarification of the same type throughout the discussion:

"The seasonal elasticity estimates specifically consider the influence of in-season precipitation on streamflow within that same season. Streamflow in many rivers is driven by out-of-season precipitation, for example, snow which falls in winter and fall may drive spring and summer streamflow as it melts, particularly in high altitude regions. Thus, while flat seasonal elasticity curves and low percentile-specific point estimates indicate a muted hydrologic response, they do not rule out the possibility that the timescale for response is merely longer than that which is considered."

This sentence, on lines 342-344 already tries to convey this point, "The range of type B elasticity curves which is present across the seasons is washed out at the annual scale, demonstrating that the catchment storage which leads to a uniform response across the distribution of streamflow generally operates at a timescale of less than year," but seems to fail at this objective, so we will try to further clarify our intended meaning here.

7. EDITING FOR EASIER COMPREHENSION: Overall the text of the paper is admirably concise, but it's still very dense. I suggest the authors review the existing text specifically to try to make it easier to

comprehend. One technique that might help is the use of tables since these facilitate a visual structure to the information. This is a suggestion only - I understand that this can be time-consuming to do!

We are happy to edit the text to try to simplify the flow of the paper to improve clarity. However, it is not immediately apparent to us how we could incorporate tables in a manner which contributes towards this goal.

ADDITIONAL MINOR COMMENTS, BY LINE:

Figure 1b: I think there's potential for confusion here: the line is monotonic increasing, but the unfamiliar reader might wonder whether it *must* be so, or whether it could be different. For example, the median (purple) point here could have higher elasticty than both the high and low flow, yes? If so, consider changing the figure to a non-monotic relationship to make it clear this is a possibility.

This is a good point and is accurate. We propose changing the figure to include a non-monotonically trending line in order to better reflect this.

Line 111-12: "We estimated ... catchment boundary". Perhaps rephrase for better clarity.

We will replace this sentence with: "We estimated average daily precipitation (mm/-

day) annually and for each season, averaged within the upstream drainage area (watershed boundary) of each gaging station."

Line 116: Unclear. Does "we recalculated these values in order to accurately represent the time period of the analysis" mean "we recalculated these because the existing dataset did not cover our desired period"?

This is correct, we will change the wording to reflect this.

Line 118: Would the sentence "Annual values..." fit better after the sentence that follows it? Also "fall into corresponding "years"" - is "water years" the intention here?

We will change this to say: "Annual values were calculated for water years (defined here as September to August), and seasonal values were estimated for winter (December, January, February), spring (March, April, May), summer (June, July, August) and fall (September, October, November) within each water year."

Figure 3: Unclear that column c is the same as Fig 2, partly because it is named differently. I suggest to name it exactly the same previously ("clusters") and potential add the words "(from Figure 2)" to make this explicit. I realise the current title is aiming to clarify normalised versus non-normalised but I feel this clarification can occur in the Figure 3 caption.

This is a good suggestion and we will implement this change.

Line 365: The introduction of example catchments interrupts the flow of the paper. Could this be done in the methods section instead?

Yes, we will introduce the example catchments in the methodology section instead and work to improve the flow of the paper.

References

Anderson, B. J., Slater, L. J., Dadson, S. J., Blum, A. G., & Prosdocimi, I. (2022). Statistical Attribution of the Influence of Urban and Tree Cover Change on Streamflow: A Comparison of Large Sample Statistical Approaches. *Water Resources Research*, *58*(5), e2021WR030742. https://doi.org/10.1029/2021WR030742

Blum, A. G., Ferraro, P. J., Archfield, S. A., & Ryberg, K. R. (2020). Causal Effect of Impervious Cover on Annual Flood Magnitude for the United States. *Geophysical Research Letters*, *47*(5), e2019GL086480. https://doi.org/10.1029/2019GL086480

Ferraro, P. J., Sanchirico, J. N., & Smith, M. D. (2019). Causal inference in coupled human and natural systems. *Proceedings of the National Academy of Sciences*, *116*(12), 5311–5318. https://doi.org/10.1073/pnas.1805563115

Imai, K., & Kim, I. S. (2021). On the Use of Two-Way Fixed Effects Regression Models for Causal Inference with Panel Data. *Political Analysis*, *29*(3), 405–415. https://doi.org/10.1017/pan.2020.33

Nichols, A. (2007). Causal Inference with Observational Data. *The Stata Journal*, *7*(4), 507–541. https://doi.org/10.1177/1536867X0800700403

Pearl, J. (2009). Causal inference in statistics: An overview. *Statistics Surveys*, *3*(0), 96–146. https://doi.org/10.1214/09-SS057

Yang, W., Yang, H., Yang, D., & Hou, A. (2021). Causal effects of dams and land cover changes on flood changes in mainland China. *Hydrology and Earth System Sciences*, *25*(5), 2705–2720. https://doi.org/10.5194/hess-25-2705-2021

---

## Author Comment (AC2)

Comments/Text of anonymous reviewer 2 posted in black, our text in blue with additions to existing text in red.

Review comments on "Elasticity curves describe streamflow sensitivity to precipitation across the entire flow distribution" by Anderson et al.

**General Comments**

This manuscript proposes a new concept in which streamflow elasticity is estimated across the full range of streamflow percentiles in a large-sample context, which is called "elasticity curve" by authors. The aim is to develop a more complete depiction of how streamflow responds to precipitation. They find three different elasticity curve types which characterize this relationship at the annual and seasonal timescales in the USA, based on two statistical modelling approaches, a panel regression which facilitates causal inference and a single catchment model which allows for consideration of static attributes. The idea is novelty and fits well with aims and objectives of HESS. This was why I accepted the review invitation. However, there are significant shortcomings in current version so that I have to recommend a rejection (below specific comments for detail).

We thank the reviewer for taking the time to read the manuscript and for clearly articulating their concern with our work. When reading their responses and criticism, we realized that the methodology was not explained well enough, which has led to major misunderstandings about our approach. Principally, the way in which Reviewer 2 has interpreted equation 2 in the manuscript is incorrect and does not represent how we have calculated elasticity for individual streamflow percentiles.

In fact, the particular way in which they have misinterpreted the method matches a point which Reviewer 1 suggested might be misunderstood. In their separate comment, Reviewer 1 stated: "I feel there is a strong possibility of readers misunderstanding the method. Specifically, the focus on different flow percentiles (or ranges of percentiles) may lead readers to believe that the method only focusses on precipitation that falls during the relevant percentile/range." This is precisely what has happened here. To clarify, mean daily precipitation and mean daily potential evaporation (calculated either annually or seasonally) is used for **all streamflow percentiles**, allowing us to develop a picture of how streamflow in different periods is affected by changes in the total precipitation. For this reason, we feel that this is a result of a confusingly worded description of the method in our manuscript and propose one primary change based on their comments.

Our proposal is to add the following clarifying statement, and to include similar examples throughout the manuscript, in order to make this clear: "As presented in this study, the elasticity curve characterizes the sensitivity of different percentiles of annual and seasonal streamflow to changes in the average annual or seasonal precipitation. For example, an elasticity of 0.5 for the 15th percentile of annual streamflow would indicate that a 1% change in the overall mean annual precipitation would correspond to a 0.5% change in the 15th percentile of annual flow."

In addition, we will adjust the text describing equation 2, so that the description of the model parameters is less ambiguous. This text is included in our response to the specific comment below. We completely understand that new methodologies can be difficult and confusing, so deeply appreciate the clarity with which Reviewer 2 has presented their concerns, as this affords us the opportunity to improve how we present our novel and useful concept.

We address each of the reviewer's specific concerns below in greater detail, and hope we have sufficiently clarified the misunderstanding so that the manuscript can be more fairly reviewed.

We have split this into 2 responses because there are 2 main critiques: 1. That we have misunderstood the concept of elasticity, and 2. concerns about the temporal distribution of the climate variables as they relate to equation 2 in the manuscript.

**Specific Comments**

First of all, authors clearly misunderstand the concept of elasticity precipitation of streamflow proposed by Schaake (1990) and Sankarasubramanian et al. (2001). The original formula is as:

$$\epsilon_{p}(P,Q) = \frac{dQ/Q}{dP/P} = \frac{dQ}{dP}\frac{P}{Q}$$
(1)

However, the difficulty with this elasticity is that we never really know dQ/dP, which is often estimated from a hydrological model and, of course, the form of the hydrological model is always unknown and validation of such a model remains a fundamental challenge (Sankarasubramanian et al. 2001; Fu et al., 2007).

In order to solve this problem, Sankarasubramanian et al. [2001] introduced a specific case of (1) at the mean value of the climatic variable:

$$\epsilon_{p}\left(\mu_{p},\mu_{Q}\right) = \frac{dQ}{dP}|_{P=\mu_{P}}\frac{\mu_{P}}{\mu_{Q}} \tag{2}$$

They (Sankarasubramanian et al. 2001 further verified that the non-parametric estimator:

$$e_{P} = median\left(\frac{Q_{t} - \overline{Q}\,\overline{P}}{P_{t} - \overline{P}\,\overline{\overline{Q}}}\right) \tag{3}$$

is a robust estimator of the precipitation elasticity of streamflow for a wide class of hydrological models that does not depend on the form of the hydrological model. This is the formula that has been wildly used in the literature to estimate the precipitation elasticity of streamflow. That is to say, the elasticity is the median value of ratio of annual streamflow anomaly in terms of long-term means to precipitation anomaly, not the long-term mean for the 50th percentile of streamflow as author claimed.

I do understand that there are some exceptions in the literature not to take this median value. For example, the two-parameter elasticity to include temperature (Fu et al., 2007) is to plot every annual ratio is plotted in a 2-d space or fitted a linear regression with these two anomalies (Zheng et al., 2009).

$$e_{P,\delta T} = \left(\frac{Q_{P,\delta T} - \overline{Q}\,\overline{P}}{P_{P,\delta T} - \overline{P}\,\overline{\overline{Q}}}\right) \tag{4}$$

$$\Delta Q_i / \overline{Q} = \epsilon \cdot \Delta X_i / \overline{X}. \tag{8}$$

We thank the reviewer for this detailed comment and explanation. We are familiar with this section of the literature on elasticity in hydrology. However, after revisiting the Sankarasubramanian et al. (2001) paper, we acknowledge that they do, in fact, estimate elasticity relative to the annual mean, rather than the median, as the reviewer points out and that the median of the ratio is taken, rather than the mean, as we originally stated. The second half of this (taking the median of the ratio, rather than the mean), was an error on our part. However, respective to the first half of this, we were explicitly referencing the elasticity of the median flow here and thus used a slightly different notation to do so. We acknowledge that this choice could have been expressed, and cited, more clearly. Regardless, we feel that this point is of little relevance to the method which is applied in the manuscript and do not believe that it implies that we have failed to understand the definition of elasticity, as the reviewer suggests.

What is of greater relevance is the abundance of literature in recent years which uses alternative methodologies, in particular, bivariate and multivariate regression-based approaches, to estimate elasticity in hydrology (for example: Andréassian et al., 2016; Bassiouni et al., 2016; Cooper et al., 2018; Potter et al., 2011; Tsai, 2017; Zhang et al., 2022, 2023). These types of approaches are often functionally equivalent (e.g. Cooper et al., 2018), or achieve a similar, or often better, result to the reference methods which are mentioned by the reviewer, as is demonstrated in Andréassian et al., 2016. Our approach, definition, and application to calculating the percentile-specific point estimates is consistent with hydrological literature in the past 15 years, as well as with the abundance of broader literature on the concept, in addition to presenting some novel additions. For the above cited reasons, we feel that this critique is of minor significance. We propose a simple modification to the manuscript in order to correct our mistake.

We have concluded that retaining the original formal definition does not add value to the paper and propose removing equation (1), where we briefly describe the Schaake (1990) and Sankarasubramanian et al. (2001) definition of elasticity, from the paper. We will change that section to read:

Historically, streamflow elasticity has been estimated using a reference approach as proposed initially by Schaake (1990) and further developed into a nonparametric estimator by Sankarasubramanian et al. (2001), in which elasticity is expressed as the median of the ratio of the annual streamflow anomaly to precipitation anomaly, relative to the long term mean. Many recent studies have instead relied on the coefficients from multivariate regression models, such as generalized and ordinary least squares regression (Andréassian et al., 2016; Potter et al., 2011), or regionally-constructed panel regression models (Bassiouni et al., 2016), to estimate elasticity. These types of approaches are often functionally equivalent (e.g. Cooper et al., 2018), or achieve a similar, or often better, result to the reference approaches, as is demonstrated in Andréassian et al., 2016. The benefits of regression-based approaches include simultaneous estimation of sensitivity to potential evaporation and precipitation, accounting for co-variation in these phenomena and providing a more robust

estimate of elasticity (Andréassian et al., 2016). Probabilistic statistical tools also enable straightforward calculation of confidence intervals.

My main scientific concern is Eq 2 of the manuscript, which is the base of this study. This does not make any scientific sense, because the same percentile of streamflow and precipitation could happen in different time of year. For example, 95th percentile of streamflow is located in June and 95th percentiles of precipitation/PET could be in December.

How possible to build a regression model between them?

$$\ln(Q_{i,t}^q) = \alpha_{i,t} + \varepsilon_P^q \ln(P_{i,t}) + \varepsilon_E^q \ln(E_{i,t}) + \eta_{i,t}^q$$

In addition, this approach requires non-zero daily streamflow for the entire study period, i.e., it cannot be applied to ephemeral rivers and streams, which limits its applications. I am surprised that it includes some rivers in Nevada and Arizona states where the number of rainfall days in a year is only 30-60 days. How can it result in a non-zero streamflow days?

We completely agree that this approach would be problematic and scientifically invalid. Fortunately, this is not what we have done!

As it is currently written, in the original submission of the manuscript, this is the text beneath equation 2 (cited above):

"where  $Q_{i,t}^q$  is the natural logarithm of a streamflow percentile (q) calculated for time period (t) for catchment (i),  $\alpha_{i,t}$  is the intercept,  $\ln(P_{i,t})$  is the logarithm of catchment averaged daily precipitation, and  $\ln(E_{i,t})$  is the logarithm of catchment averaged daily potential evaporation. The point estimate of precipitation elasticity is represented by the regression coefficient:  $\varepsilon_p^q$ and potential evaporation elasticity is represented by  $\varepsilon_E^q$  The error term is  $\eta_{i,t}^q$ ."

A close examination of the text and the associated equation will reveal that the superscript "q" is not present for the variables  $\ln(P_{i,t})$  or  $\ln(E_{i,t})$ . This indicates that the variables are not specific percentiles, but rather the "catchment averaged daily [variable]" as described in the text.

However, we acknowledge that the wording here was ambiguous and can completely understand why this would be confusing. To rectify this, we propose adding the following to the text (red) in the interest of clarifying this:

"where  $Q_{i,t}^q$  is the natural logarithm of a streamflow percentile (q) calculated for time period (t) for catchment (i),  $\alpha_{i,t}$  is the intercept,  $\ln(P_{i,t})$  is the logarithm of catchment averaged mean daily precipitation for the time period of interest (year or season), and  $\ln(E_{i,t})$  is the logarithm of catchment averaged mean daily potential evaporation in that period. Note that mean seasonal and annual climate time series are used, not percentiles equivalent to the streamflow percentile of interest (denoted with the superscript "q"). The point estimate of precipitation elasticity is represented by the regression coefficient:  $\varepsilon_p^q$  and potential evaporation elasticity is represented by  $\varepsilon_p^q$ . The error term is  $\eta_{it}^q$ ."

In addition, we will include clarifying examples as suggested by Reviewer 1 and described in the overview of changes above.

**Reference**

Fu, G., S.P. Charles, F.H.S. Chiew (2007), A two-parameter climate elasticity of streamflow index to assess climate change effects on annual streamflow, Water Resour. Res., 43 (2007), p. W11419, 10.1029/2007WR00589

Sankarasubramanian, A., R. M. Vogel, and J. F. Limbrunner (2001), Climate elasticity of streamflow in the United States, Water Resour. Res., 37, 1771–1781.

Schaake, J. C. (1990), From climate to flow, in Climate Change and U.S. Water Resources, edited by P.E. Waggoner, chap. 8, pp. 177–206, John Wiley, New York.

Zheng, H., L. Zhang, R. Zhu, C. Liu, Y. Sato, Y. Fukushima (2009), Responses of streamflow to climate and land surface change in the headwaters of the Yellow River Basin. Water Resour. Res. 45, W00A19.

Andréassian, V., Coron, L., Lerat, J., & Le Moine, N. (2016). Climate elasticity of streamflow revisited – an elasticity index based on long-term hydrometeorological records. Hydrology and Earth System Sciences, 20(11), 4503–4524. https://doi.org/10.5194/hess-20-4503-2016

Bassiouni, M., Vogel, R. M., & Archfield, S. A. (2016). Panel regressions to estimate low-flow response to rainfall variability in ungaged basins. Water Resources Research, 52(12), 9470–9494. https://doi.org/10.1002/2016WR018718

Cooper, M. G., Schaperow, J. R., Cooley, S. W., Alam, S., Smith, L. C., & Lettenmaier, D. P. (2018). Climate Elasticity of Low Flows in the Maritime Western U.S. Mountains. Water Resources Research, 54(8), 5602–5619. https://doi.org/10.1029/2018WR022816

Potter, N. J., Petheram, C., & Zhang, L. (2011). Sensitivity of streamflow to rainfall and temperature in south-eastern Australia during the Millennium drought. 19th International Congress on Modelling and Simulation, Perth, Dec, 3636–3642.

Tsai, Y. (2017). The multivariate climatic and anthropogenic elasticity of streamflow in the Eastern United States. Journal of Hydrology: Regional Studies, 9, 199–215. https://doi.org/10.1016/j.ejrh.2016.12.078

Zhang, Y., Viglione, A., & Blöschl, G. (2022). Temporal Scaling of Streamflow Elasticity to Precipitation: A Global Analysis. Water Resources Research, 58(1), e2021WR030601. https://doi.org/10.1029/2021WR030601

Zhang, Y., Zheng, H., Zhang, X., Leung, L. R., Liu, C., Zheng, C., Guo, Y., Chiew, F. H. S., Post, D., Kong, D., Beck, H. E., Li, C., & Blöschl, G. (2023). Future global streamflow declines are probably more severe than previously estimated. Nature Water, 1–11. https://doi.org/10.1038/s44221-023-00030-7

---

## Author Response (AR2)

Overall, the authors have been responsive to reviewer suggestions, and the manuscript has improved relative to the first version. The authors have clarified several sections of the manuscript, particularly its framing, and have also moved material to the Appendix, which has streamlined the article favourably.

I maintain my opinion that the research is of high quality and is worthy of publication in HESS. In my opinion, the article is almost ready to be finalised. I have only a few suggestions of improvement, as below.

We thank the reviewer for their comments, both in this and the previous round of revisions. These have greatly improved the clarity of the paper, for which we are grateful. We address each individual comment below.

Firstly, in line 17 of the updated abstract (which is much improved!) I suggest to delete or alter the words "in a large-sample context". The danger is that the reader interprets this incorrectly to mean that the method is only applicable to multiple catchments. In practice, it is possible to apply it to one catchment only, if desired.

We have deleted the phrase as suggested.

Secondly, my earlier suggestion was left unanswered, namely I said:

"the authors ought to aim to clearly establish ... the need for the new method, couched in terms of the limitations of the existing method. [The manuscript] does not go into sufficient detail (saying only "abnormally high and low flows are associated with the greatest strain on hydrological systems"). Can we get a lot more detail here? Eg. for high flows, it could acknowledge/discuss that infrastructure is often designed according to estimates of flooding potential, so any changes to this potential are very important; likewise low flows are important eg. for riverine ecology among other things. Articulating these factors will help the reader understand why the new method is important, which will motivate them to keep reading."

I do feel it is always important to make the strongest possible statement of the relevance/importance of a manuscript to real-world outcomes. Thus, I would ask the authors to consider this again. I feel this would require only, say, three or four more sentences in the introduction. I note that the authors have added text clarifying the hydrology of low flows and high flows (a nice addition) but this is not the same as stating their importance to societal problems and/or to related fields of science such as aquatic ecology.

This is fair and we do wish for the broad relevance and applicability of this work to be clear. We have added the following to the introduction:

"Understanding the sensitivity of each of these components of the flow regime is important considering their unique roles in determining resilience and adaptability to climatic change. For instance, low flows are highly relevant for riverine ecology, water quality, and water availability for out-of-channel water uses like irrigation, power generation, and municipal water supply (Cooper et al., 2018; Smakhtin, 2001). High flows correspond to flooding, and understanding their distributions and probability is essential for flood frequency estimation and infrastructure planning, among other things (François et al., 2019). The typical approaches of estimating elasticity for a single point along the flow distribution are insufficient for the objective of characterizing flow response to precipitation change since the elasticity of the central summary of the distribution is unlikely to capture hydrologic behaviour in either low or high flow percentiles. "

Thirdly, I would just ask the authors to do a quick check to ensure all the edits they have described are in the final manuscript. For example, the extra words they suggest for the caption of Figure 1b, starting with the words "Note: In practice...", are not in the new version. I am confident this is merely an oversight, so a quick check should be sufficient to catch any similar omissions.

We have corrected this oversight and have validated that each of our other suggested changes were made with the previous revisions.

My compliments to the authors for this interesting and relevant research.

Thank you very much for the complimentary and constructive reviews!

Keirnan Fowler